# Synthesis, Antifungal Activity, 3D-QSAR and Controlled Release on Hydrotalcite Study of Longifolene-Derived Diphenyl Ether Carboxylic Acid Compounds

**DOI:** 10.3390/molecules28041911

**Published:** 2023-02-16

**Authors:** Xiaocui Wu, Guishan Lin, Wengui Duan, Baoyu Li, Yucheng Cui, Bo Cen, Fuhou Lei

**Affiliations:** 1School of Chemistry and Chemical Engineering, Guangxi University, Nanning 530004, China; 2Guangxi Key Laboratory of Chemistry and Engineering of Forest Products, Guangxi Collaborative Innovation Center for Chemistry and Engineering of Forest Products, Guangxi Minzu University, Nanning 530006, China

**Keywords:** longifolene, hydrotalcite, antifungal activity, 3D-QSAR, controlled-releasing

## Abstract

Twenty-two novel longifolene-derived diphenyl ether-carboxylic acid compounds **7a–7v** were synthesized from renewable biomass resources longifolene, and their structures were confirmed by FT-IR, ^1^H NMR, ^13^C NMR, and HRMS. The preliminary evaluation of in vitro antifungal activity displayed that compound **7b** presented inhibition rates of 85.9%, 82.7%, 82.7%, and 81.4% against *Alternaria solani, Cercospora arachidicola*, *Rhizoctonia solani,* and *Physalospora piricola*, respectively, and compound **7l** possessed inhibition rates of 80.7%, 80.4%, and 80.3% against *R. solani*, *C. arachidicola*, *P. piricola*, respectively, exhibiting excellent and broad-spectrum antifungal activities. Besides, compounds **7f** and **7a** showed significant antifungal activities with inhibition rates of 81.2% and 80.7% against *A.solani*, respectively. Meanwhile, a reasonable and effective 3D-QSAR mode (*r*^2^ = 0.996, *q*^2^ = 0.572) has been established by the CoMFA method. Furthermore, the drug-loading complexes **7b/MgAl-LDH** were prepared and characterized. Their pH-responsive controlled-release behavior was investigated as well. As a result, complex **7b/MgAl-LDH-2** exhibited excellent controlled-releasing performance in the water/ethanol (10:1, v:v) and under a pH of 5.7.

## 1. Introduction

Fungicides have been extensively used in controlling plant diseases around the world, playing a critical role in the protection of crops [1,2,3]. However, the increasingly serious resistance to current commercial fungicides in long-term use and the low utilization efficiency due to the loss of active ingredients resulting from volatilization and decomposition etc., are the main problems that need to be addressed [4,5]. It is, therefore, needful to develop novel fungicide candidates to overcome the increasingly serious resistance and more effective drug-delivery systems to improve utilization efficiency.

A prospective approach to resolve the low utilization efficiency is the fabrication of controlled-release pesticide systems to prevent volatilization and decomposition and to extend the duration [6,7]. In recent years, controlled-release nanopesticides, as one of the major methods of pesticide development, have attracted increasing attention all over the world [8,9,10,11]. A variety of nanomaterials have served as carriers in those controlled-release pesticide systems [12,13,14]. However, layered double hydroxides (LDHs), a nanomaterial with the formula expressed as [M_1 − *x*_^2+^M*_x_*^3+^(OH)_2_] (A*^n^*_−_)*_x_*_/*n*_·*m*H_2_O, are made up of positively charged metal host layers and exchangeable interlayer anions [15,16]. Because of low toxicity [17], excellent biocompatibility [18], UV protection performance [19], and the distinct structure with tunability in both host layer and interlayer anions [20], LDHs have revealed potential value for application as excellent drug carriers [21,22]. Nevertheless, it was scarcely investigated for pesticides [23,24].

On the other hand, heavy turpentine oil is a byproduct in the production of turpentine and rosin from living pine trees but was just applied to an inexpensive boiler fuel [25,26]. Its main component is tricyclic sesquiterpene longifolene. To improve the value-added application of these green forest resources, some compounds with good biological activities have been developed by Wang and our research teams from longifolene or isolongifolene, which was converted by the isomerization from longifolene for the past few years [27,28,29,30,31,32]. 

In our previous work, the phenolic acid derivative methyl 4-(2-hydroxy-5-isopropyl phenyl)-4-methyl pentanoate was prepared from longifolene [28]. According to the structural requirement of the exchangeable interlayer anion of LDHs [33], in this work, this phenolic acid derivative was converted into a series of longifolene-derived diphenyl ether carboxylic acid derivatives. It was used as the potentially exchangeable interlayer anion with biological activity in light of the good exhibition of diphenyl ether derivatives in the development of pesticides, which were employed as pesticides such as the herbicide Bifenox, the insecticide Pyriproxyfen and the fungicide Metominostrobin, and were continuously developed for crop protection [34,35,36,37]. All the target compounds were characterized and preliminarily evaluated for in vitro antifungal activities against eight fungi. And a preliminary three-dimensional quantitative structure-activity relationship (3D-QSAR) model was built by the comparative molecular field analysis (CoMFA) method. Then, the longifolene-derived diphenyl ether carboxylic acid/nano-hydrotalcite complexes were prepared by loading the screened compound **7b** with the best antifungal activity onto **MgAl-LDH** nanosheets and characterized, drug-loading complexes **7b/MgAl-LDH** pH-responsive controlled-release behavior was investigated as well.

## 2. Results and Discussion

### 2.1. Synthesis and Characterization

As illustrated in Figure 1, compound **2** was obtained by isomerization- aromatization reaction of sustainable biomass resource longifolene and further oxidized by TBHP oxidant to prepare compound **3**. Then, compound **4** was obtained by the Baeyer-Villager rearrangement using *m*-CPBA as the oxidant and converted to compound **5**. The longifolene-derived diphenyl ether-carboxylic methyl ester compounds **6a–6v** were generated by coupling. Lastly, a series of longifolene-derived diphenyl ether-carboxylic acid compounds **7a–7v** were synthesized by hydrolysis.

The structures of all the compounds were identified by FT-IR, ^1^H NMR, ^13^C NMR, and HRMS. In the IR spectra, the characteristic absorption bands at about 3091−3011 and 1711−1701 cm^−1^ were assigned to the stretching vibrations of the Ar−H and C=O, respectively. The ^1^H NMR spectra of **7a–7v** showed characteristic signals at δ 6.53−7.44 ppm, which were attributed to the protons of the benzene ring. The methylene protons bonded to the benzene ring displayed signals at about 2.87 ppm. And the characteristic signals at about 2.12–2.18 ppm were assigned to the methylene protons on the saturated carbon bonded to the carbonyl carbon atom. The other protons bonded to the saturated carbons displayed signals in the range of δ 1.24–1.40 ppm. The ^13^C NMR spectra of all the target compounds exhibited peaks for the carbons of C=O on the carboxylic acid at about δ 175.33 ppm and carbon atoms of the benzene ring at 118.56–157.49 ppm. The other saturated carbons displayed signals in the region of 24.20–37.69. Their molecular weights were in accordance with the consequences of HRMS. Besides, all the related characterization data and spectra above can be found in the Appendix A. 

### 2.2. Antifungal Activity

The antifungal activities of the target compounds **7a–7v** were evaluated by the in vitro method against eight plant pathogens at 50 μg/mL [38], including cucumber fusarium wilt (*Fusarium oxysporum f. sp. Cucumerinum*), peanut speckle (*Cercospora arachidicola*), apple root spot (*Physalospora piricola*), tomato early blight (*Alternaria solani*), wheat scab (*Gibberella zeae*), rice sheath blight (*Rhizoctonia solani*), corn southern leaf blight (*Bipolaris maydis*), watermelon anthracnose (*Colletotrichum osrbiculare*). The commercial fungicide chlorothalonil was used as the positive control. The results are listed in Table 1.

Most of the compounds showed certain antifungal activities against the tested fungi. Among them, compound **7b** presented inhibition rates of 85.9%, 82.7%, 82.7%, and 81.4% against *A. solani*, *C. arachidicola*, *R. solani*, and *P. piricola*, respectively, and compound **7l** possessed inhibition rates of 80.7%, 80.4%, and 80.3% against *R. solani*, *C. arachidicola*, *P. piricola*, respectively, exhibiting excellent and broad-spectrum antifungal activities. Besides, compounds **7f** and **7a** showed significant antifungal activities with inhibition rates of 81.2% and 80.7% against *A. solani*, respectively. It was also found that the substituent group R presented an obvious influence on antifungal activity, and a 3D-QSAR study was subsequently performed.

### 2.3. 3D-QSAR Analysis

3D-QSAR analysis of the experimental and predicted antifungal activity against *A. solani* for the target compounds was carried out by the CoMFA method. The experimental and predicted activities of the training set are presented in Table 2, a predictive 3D-QSAR model with the conventional correlation coefficient *r*^2^ = 0.996 and the cross-validated coefficient *q*^2^ = 0.572 is shown in Table 3. As presented in Figure 1, the scatter plot of the predicted active factor (AF) values versus experimental AF values is shown. The whole data converged near the X = Y line, implying that the 3D-QSAR model was credible and had a nice predictive ability.

The steric and electrostatic field contour maps of CoMFA were demonstrated in Figure 2. The contribution rate for steric and electrostatic fields were 64.8% and 35.2% (Table 3), respectively, manifesting the steric field was the major benefit to the improvement of the antifungal activity against *A.solani*. As shown in Figure 2a, there was a multitude of blue areas around the 3-position of the benzene ring and red areas suspended above the 4-position of the benzene ring. The blue region represented that the introduction of electron-donating groups was conducive to increase activity, and the red region expressed that the introduction of electron-withdrawing groups was conducive to enhancing activity. For instance, compound **7e** (R = 3-CH_3_) showed a higher inhibitory rate than that of **7l** (R = 3-CN) and **7n** (R = 3-F), and compound **7b** (R = 4-CN) revealed preferable antifungal activity than **7i** (R = 4-OCH_3_) and **7j** (R = 4-CH_3_). Furthermore, in Figure 2b, there were a few green areas distributed around the 2-position or 4-position of the benzene ring, and the green region shows that the introduction of large groups is beneficial to perfect the antifungal activity. For example, compounds **7h** (R = 4-Ph) exhibited better antifungal activity than **7j** (R = 4-CH_3_). Based on the results of the 3D-QSAR analysis above, a new compound (Figure 3) was designed, and its AF was predicted by the established CoMFA model. As a result, the predicted AF was high to −1.71, showing excellent antifungal activity, which needs to be further verified by experiment.

### 2.4. Preparation and Characterization of **MgAl-LDH** and **7b/MgAl-LDH**

Generally, **MgAl-LDH** was synthesized by a “bottom-up” method [21,39], and drug-loading complexes **7b/MgAl-LDH** were fabricated by the self-assembly method of carriers **MgAl-LDH** and bioactive compound **7b**. As shown in Figure 4a, the XRD pattern of **MgAl-LDH** displayed characteristic diffraction peaks at 2*θ* = 11.3°, 22.8°, and 34.4° (Figure 4a red line), corresponding to 003, 006, and 009 lattices of the **MgAl-LDH** structure. Compared with that of **MgAl-LDH**, there were several new diffraction peaks at 2*θ* = 4.9°, 30.2°, and 44.5° (Figure 4a; blue line), in the sample **7b/MgAl-LDH-2**, and the diffraction peaks at 2*θ* = 11.3°, 22.8°, and 34.4° became more narrow. It was suggested that the crystalline region of nano-hydrotalcite was destructed during the formation of complex **7b/MgAl-LDH-2**. Then, the FT-IR spectra of bioactive compound **7b**, carrier **MgAl-LDH** and drug-loading complex **7b/MgAl-LDH-2** were shown in Figure 4b. The absorption bands at 1710, 2225, and 3450 cm^−1^ were attributed to the stretching vibration of C=O, CN, and COOH of compound **7b**, respectively. The characteristic vibration absorption peak at 1367 cm^−1^ attributed to NO_3_^-^ of MgAl-LDH was also observed in that of complex **7b/MgAl-LDH-2**, indicating the success in the formation of complex **7b/MgAl-LDH**. 

As shown in Figure 4c,d, the atomic force microscopy (AFM) images of **MgAl-LDH** and **7b/MgAl-LDH-2** exhibited their sheet-like structures and compared with that of **MgAl-LDH** (~1.1 nm), the thickness of **7b/MgAl-LDH-2** was bigger (~1.3 nm), which was due to the combination of the small-molecule drug (compound **7b**) and carrier (**MgAl-LDH**).

### 2.5. Micro-Morphologies and In Vitro pH Controlled-Releasing Properties of Drug-Loading Complexes

To further clarify the microstructures of the newly prepared drug-loading complex **7b/MgAl-LDH-2**, the SEM images of complex **7b/MgAl-LDH-2** were taken in comparison with those of carrier **MgAl-LDH**. As shown in Figure 5, it was found that the shape of complex **7b/MgAl-LDH-2** was similar to that of **MgAl-LDH** nanosheets, indicating that the particles of compound **7b** were intercalated to the inner of **MgAl-LDH** nanosheets, and uniformly distributed at the inner and surface of **MgAl-LDH** nanosheets. In addition, the aggregation of the nanoparticles of drug-loading complex **7b/MgAl-LDH-2** led to compact and massive structures, which were favorable to its pH-responsive controlled-releasing performance.

Then, the in vitro releasing behaviors of drug-loading complexes **7b/MgAl-LDH-1**, **7b/MgAl-LDH-2** and **7b/MgAl-LDH-3** in EtOH-H_2_O solution (1:10, v/v) with different pH values (5.7, 7.0, and 8.2) was investigated at room temperature. As shown in Figure 6, all of the tested samples showed sustained releasing properties in the specific conditions, but their releasing rates and total releasing amount were different. For all three drug-loading complexes, the releases of the bioactive compound into alkaline and neutral conditions were relatively difficult; only 25.6% and 48.9% of the total loaded compound **7b** was released from complexes **7b/MgAl-LDH-2**, respectively. In contrast, 92.4% of compound **7b** was released after 400 h at pH = 5.7. In acidic conditions, the three drug-loading complexes displayed significant, sustained releasing properties. In addition, the complexes **7b/MgAl-LDH-2** showed obvious releasing properties compared with **7b/MgAl-LDH-1** and **7b/MgAl-LDH-3**. Therefore, under different pH conditions, the cases of drug release of these complexes were different, showing that the pH value of the drug-releasing environment could regulate and control the drug-releasing pattern of the corresponding complexes.

## 3. Experimental Section

### 3.1. Materials

Longifolene (GC purity 85%) (**1**) was obtained from Wuzhou Pine Chemicals Co., Ltd., Wuzhou, China. Zinc chloride (98%), tert-butyl hydroperoxide solution (TBHP) (70%), 1-(Pyridin-2-yl)propan-2-one (95%), and 3-chloroperoxybenzoic acid (85%) were purchased from Macklin Reagents (Shanghai). Dimethyl sulfoxide (99%), methanol (99%), acetonitrile (99%), dichloromethane (99%), and tetrahydrofuran (THF, 99%) were obtained from Sinopharm Chemicals Reagents Co., Ltd. (Shanghai, China). Acetyl chloride (99%) and cesium carbonate (99%) were acquired from Aladdin Reagents (Shanghai, China). Sodium hydroxide (98%) and triethylamine (98%) were purchased from Guanghua Sci-Tech Co., Ltd. (Guangdong, China). All other reagents and solvents used in our work were commercially available and used without further purification.

### 3.2. Chemical Synthesis

As illustrated in Figure 1, compound **2**, yielding 55.8%, was obtained by isomerization- aromatization reaction of sustainable biomass resource longifolene and further oxidized by TBHP oxidant to prepare compound **3**, yielding 69.5%. Then, compound **4**, yielding 78.8%, was obtained by the Baeyer-Villager rearrangement using *m*-CPBA as the oxidant. Compounds **2–4** were synthesized in accordance with the methods in our previous studies [27,28].

Synthesis of the phenolic acid derivative methyl 4-(2-hydroxy-5-isopropyl phenyl)-4-methyl pentanoate (**5**). To a mixture of compound **4** (8.2 g, 35 mmol) and dry methanol (30 mL), a catalyst of 80 μL H_2_SO_4_ was added. Then, the reaction mixture was refluxed for 6 h. After the mixture was extracted with diethyl ether (3 × 40 mL), the combined organic phase was dried over anhydrous Na_2_SO_4_ and evaporated the solvent in a vacuum. Finally, the residue was further purified by column chromatography on silica gel with petroleum ether and EtOAc (20:1, v/v) as the eluent to gain a white solid, compound **5**, at an 89.7% yield.

General procedure for the synthesis of longifolene-derived diphenyl ether-carboxylic methyl ester compounds **6a–6v**. In a dry sealed tube was charged with a magnetic stir bar, CuBr (7 mg, 0.05 mmol), Cs_2_CO_3_ (650 mg, 2 mmol), compound **5** (0.32 g, 1 mmol), and 1.2 mmol of different halides (if solids). Drain the tube and backfill it with nitrogen (repeat this process three times). 1.2 mmol of different halides (if liquid), DMSO (1.5 mL), and 2-picolyl methyl ketone (14 mg, 0.1 mmol) were added with a syringe under nitrogen protection. The reaction mixture was heated to the indicated temperature (90 °C) for 24 h. After cooling to room temperature, the mixture was diluted with EtOAc (10 mL), filtered out the inorganic salt, and the combined organic layer was concentrated in a vacuum. Finally, the residue was purified by column chromatography on silica gel with petroleum ether and EtOAc (10:1, v/v) as the eluent to obtain compound **6a–6v**. Yield: 53.7–86.3%.

General procedure for the synthesis of longifolene-derived diphenyl ether-carboxylic acid compounds **7a–7v**. A solution of NaOH (0.06 g, 1.5 mmol) in water (5 mL) was added to a mixture of compound **6** (0.34 g, 1.0 mmol) and methanol (30 mL) in a flask, and the resulting mixture was refluxed for 5 h. Then, the mixture was acidified with 90 μL sulfuric acid and extracted with EtOAc (3 × 30 mL). The combined organic phase was dried and evaporated. Finally, the residue was purified by column chromatography on silica gel using petroleum ether and EtOAc (5:1, v/v) as eluent to obtain the desired target compounds **7a–7v**.

Compound (**7a**): *4-(5-isopropyl-2-phenoxyphenyl)-4-methylpentanoic acid.* White solid; yield 96.4%; m. p. 110.6–111.4 °C; FT-IR (KBr, cm^−1^): 3039 (Ar–H), 2966, 2926, 2872 (CH), 1705 (C=O), 1588, 1486, 1456, 1305, 1232 (Ar-C=C); ^1^H NMR (500 MHz, CDCl_3_) δ 7.38–7.25 (m, 2H, C_13_-H, C_15_-H), 7.13 (s, 1H, C_6_-H), 7.04 (t, *J* = 7.4 Hz, 1H, C_14_-H), 6.97 (dd, *J* = 14.5, 9.1 Hz, 3H, C_8_-H C_12_-H, C_16_-H), 6.73 (d, *J* = 8.3 Hz, 1H, C_9_-H), 2.87 (dt, *J* = 13.8, 6.9 Hz, 1H, C_17_-H), 2.18 (d, *J* = 4.9 Hz, 2H, C_2_-H), 2.12 (d, *J* = 4.9 Hz, 2H, C_3_-H), 1.40 (s, 6H, C_20_-H, C_21_-H) ), 1.24 (d, *J* = 6.9 Hz, 6H, C_18_-H, C_19_-H); ^13^C NMR (126 MHz, CDCl_3_) δ 180.36, 157.49, 153.47, 143.50, 137.06, 129.64, 126.52, 124.96, 122.60, 119.79,118.56, 37.69, 35.91, 33.74, 30.56, 28.41, 24.20; HRMS *m*/*z*:325.1808 [M + H^+^].

Compound (**7b**): *4-(2-(4-cyanophenoxy)-5-isopropylphenyl)-4-methylpentanoic acid.* White solid; yield 98.4%; m. p. 143.6–144.8 °C; FT-IR (KBr, cm^−1^): 3042 (Ar–H), 2966 (C-H), 2225 (C≡N), 1707 (C=O), 1595, 1489, 1457, 1307, 1230 (Ar-C=C); ^1^H NMR (500 MHz, CDCl_3_) δ 7.58 (d, *J* = 11.5 Hz, 2H, C_13_-H, C_15_-H), 7.18 (s, 1H, C_6_-H), 7.07 (d, *J* = 10.4 Hz, 1H, C_8_-H), 6.99 (d, *J* = 9.5 Hz, 2H, C_12_-H, C_16_-H), 6.79 (d, *J* = 8.2 Hz, 1H, C_9_-H), 2.97–2.79 (m, 1H, C_17_-H), 2.09 (s, 4H, C_2_-H, C_3_-H), 1.35 (s, 6H, C_20_-H, C_21_-H)), 1.26 (d, *J* = 6.9 Hz, 6H, C_18_-H, C_19_-H); ^13^C NMR (126 MHz, CDCl_3_) δ 179.96, 161.59, 151.20, 145.53, 138.14, 134.13, 127.09, 125.45, 121.41, 118.92, 117.92, 105.45, 37.68, 36.06, 33.85, 30.38, 28.47, 24.13; HRMS *m*/*z*: 350.1760 [M + H^+^].

Compound (**7c**): *4-(2-(4-fluorophenoxy)-5-isopropylphenyl)-4-methylpentanoic acid.* White solid; yield 97.2%; m. p. 126.5–127.8 °C; FT-IR (KBr, cm^−1^): 3080 (Ar–H), 2960, 2872 (C-H), 1709 (C=O), 1504, 1488, 1457, 1293, and 1209 (Ar-C=C); ^1^H NMR (500 MHz, CDCl_3_) δ 7.12 (d, *J* = 2.1 Hz, 1H, C_6_-H), 6.98 (d, *J* = 8.2 Hz, 3H, C_8_-H, C_13_-H, C_15_-H), 6.92 (d, *J* = 6.9 Hz, 2H, C_12_-H, C_16_-H), 6.67 (d, *J* = 8.3 Hz, 1H, C_9_-H), 2.87 (dt, *J* = 13.8, 6.9 Hz, 1H, C_17_-H), 2.19 (d, *J* = 9.1 Hz, 2H, C_2_-H), 2.12 (d, *J* = 9.2 Hz, 2H, C_3_-H), and 1.40 (s, 6H, C_20_-H, C_21_-H), 1.23 (d, *J* = 6.9 Hz, 6H, C_18_-H, C_19_-H); ^13^C NMR (126 MHz, CDCl_3_) δ 180.27, 159.41, 157.50, 153.90, 153.22, 143.49, 136.76, 126.60, 125.02, 119.98, 119.09, 116.27, 116.08, 37.71, 35.88, 33.72, 30.55, 28.36, and 24.19; HRMS *m*/*z*: 343.1714 [M + H^+^].

Compound (**7d**): *4-(5-isopropyl-2-(o-tolyloxy)phenyl)-4-methylpentanoic acid.* White solid; yield 98.1%; m. p. 112.6–113.9 °C; FT-IR (KBr, cm^−1^): 3032 (Ar–H), 2954, 2923, 2869 (C-H), 1708 (C=O), 1485, 1460, 1413, 1304, 1237, and 1208 (Ar-C=C); ^1^H NMR (600 MHz, CDCl_3_) δ 7.23 (d, *J* = 7.0 Hz, 1H, C_6_-H), 7.13–7.10 (m, 2H, C_13_-H, C_15_-H), 7.00 (t, *J* = 7.8 Hz, 1H, C_14_-H), 6.94 (d, *J* = 10.4 Hz, 1H, C_8_-H), 6.79 (d, *J* = 8.0 Hz, 1H, C_9_-H), 6.54 (d, *J* = 8.3 Hz, 1H, C_12_-H), 2.86 (dt, *J* = 13.8, 6.9 Hz, 1H, C_17_-H), 2.28 (s, 3H, C_22_-H), 2.25 (d, *J* = 8.9 Hz, 2H, C_2_-H), 2.14 (dd, *J* = 9.3, 7.3 Hz, 2H, C_3_-H), and 1.44 (s, 6H, C_20_-H, C_21_-H), 1.23 (d, *J* = 6.9 Hz, 6H, C_18_-H, C_19_-H); ^13^C NMR (151 MHz, CDCl_3_) δ 180.33, 154.88, 154.04, 142.63, 131.21, 127.09, 126.51, 124.86, 123.19, 118.66, 117.62, 37.77, 35.89, 33.67, 30.64, 28.36, 24.22, and 16.35; HRMS *m*/*z*: 339.1964 [M + H^+^].

Compound (**7e**): *4-(5-isopropyl-2-(m-tolyloxy)phenyl)-4-methylpentanoic acid.* White solid; yield 95.7%; m. p. 110.5–112.0 °C; FT-IR (KBr, cm^−1^): 2961, 2871, 2869 (C-H), 1709 (C=O), 1591, 1485, 1410, 1307, 1259, and 1229 (Ar-C=C); ^1^H NMR (600 MHz, CDCl_3_) δ 7.18 (t, *J* = 7.8 Hz, 1H, C_13_-H), 7.13 (d, *J* = 2.2 Hz, 1H, C_6_-H), 6.99 (dd, *J* = 8.3, 2.1 Hz, 1H, C_8_-H), 6.86 (d, *J* = 7.5 Hz, 1H, C_16_-H), 6.80 (s, 1H, C_14_-H), 6.74 (t, *J* = 10.3 Hz, 2H, C_9_-H C_12_-H), 2.87 (dt, *J* = 13.8, 6.9 Hz, 1H, C_17_-H), 2.31 (s, 3H, C_22_-H), 2.22–2.18 (m, 2H, C_2_-H), 2.15–2.10 (m, 2H, C_3_-H), 1.41 (s, 6H, C_20_-H, C_21_-H), and 1.24 (d, *J* = 6.9 Hz, 6H, C_18_-H, C_19_-H); ^13^C NMR (151 MHz, CDCl_3_) δ 180.51, 157.40, 153.60, 143.32, 139.83, 136.94, 129.34, 126.48, 124.93, 123.48, 119.69, 119.32, 115.58, 37.69, 35.89, 33.73, 30.5, 28.40, 24.21, and 21.38; HRMS *m*/*z*: 339.1963 [M + H^+^].

Compound (**7f**): *4-(2-(4-bromophenoxy)-5-isopropylphenyl)-4-methylpentanoic acid.* White solid; yield 98.2%; m. p. 133.3–134.6 °C; FT-IR (KBr, cm^−1^): 2961, 2871, 2869 (C-H), 1708 (C=O), 1481, 1411, 1241, and 1215 (Ar-C=C); ^1^H NMR (600 MHz, CDCl_3_) δ 7.39 (d, *J* = 8.9 Hz, 2H, C_13_-H, C_15_-H), 7.13 (s, 1H, C_6_-H), 7.01 (d, *J* = 10.4 Hz, 1H, C_8_-H), 6.84 (d, *J* = 2.1 Hz, 2H, C_12_-H, C_16_-H), 6.73–6.71 (m, 1H, C_9_-H), 2.87 (dt, *J* = 13.8, 6.9 Hz, 1H, C_17_-H), 2.15 (d, *J* = 9.3 Hz, 2H, C_2_-H), 2.13–2.09 (m, 2H, C_3_-H), 1.38 (s, 6H, C_20_-H, C_21_-H), and 1.24 (d, *J* = 6.9 Hz, 6H, C_18_-H, C_19_-H); ^13^C NMR (151 MHz, CDCl_3_) δ 180.14, 156.82, 152.91, 144.14, 138.56, 137.33, 132.58, 126.71, 125.15, 120.48, 120.04, 114.94, 37.68, 35.93, 33.76, 30.48, 28.41, and 24.17; HRMS *m*/*z*: 403.0910 [M + H^+^].

Compound (**7g**): *4-(5-isopropyl-2-(4-nitrophenoxy)phenyl)-4-methylpentanoic acid.* White solid; yield 94.8%; m. p. 133.6–134.9 °C; FT-IR (KBr, cm^−1^): 2961, 2923 (C-H), 1702 (C=O), 1588, 1514, 1487, 1347, and 1253 (Ar-C=C); ^1^H NMR (600 MHz, CDCl_3_) δ 8.19 (d, *J* = 9.2 Hz, 2H, C_13_-H, C_15_-H), 7.20 (d, *J* = 2.0 Hz, 1H, C_6_-H), 7.10 (d, *J* = 8.2 Hz, 1H, C_8_-H), 7.01 (t, *J* = 6.1 Hz, 2H, C_12_-H, C_16_-H), 6.82 (d, *J* = 8.2 Hz, 1H, C_9_-H), 2.92 (dt, *J* = 13.8, 6.9 Hz, 1H, C_17_-H), 2.09 (s, 4H, C_2_-H, C_3_-H), 1.36 (s, 6H, C_20_-H, C_21_-H), and 1.27 (d, *J* = 6.9 Hz, 6H, C_18_-H, C_19_-H); ^13^C NMR (151 MHz, CDCl_3_) δ 179.96, 163.31, 151.13, 145.86, 142.47, 138.21, 127.20, 125.99, 125.57, 121.62, 117.11, 37.70, 36.03, 33.88, 30.30, 28.49, and 24.13; HRMS *m/z*: 370.1659 [M + H^+^].

Compound (**7h**): *4-(2-([1,1′-biphenyl]-4-yloxy)-5-isopropylphenyl)-4-methylpentanoic acid.* White solid; yield 96.5%; m. p. 148.8–150.2 °C; FT-IR (KBr, cm^−1^): 3030 (Ar–H), 2960, 2867 (C-H), 1705 (C=O), 1486, 1411, 1299, 1244, and 1167 (Ar-C=C); ^1^H NMR (600 MHz, CDCl_3_) δ 7.53 (dd, *J* = 17.8, 8.1 Hz, 4H, C_12_-H, C_13_-H, C_15_-H, C_16_-H), 7.40 (t, *J* = 7.6 Hz, 2H, C_19_-H, C_21_-H), 7.29 (s, 1H, C_8_-H), 7.14 (s, 1H, C_20_-H), 7.02 (d, *J* = 8.5 Hz, 3H, C_6_-H, C_18_-H, C_22_-H), 6.80 (d, *J* = 8.3 Hz, 1H, C_9_-H), 2.88 (dt, *J* = 13.8, 6.9 Hz, 1H, C_23_-H), 2.23–2.16 (m, 2H, C_2_-H), 2.14–2.08 (m, 2H, C_3_-H), 1.41 (s, 6H, C_26_-H, C_27_-H), and 1.25 (d, *J* = 6.9 Hz, 6H, C_24_-H, C_25_-H); ^13^C NMR (151 MHz, CDCl_3_) δ 180.47, 157.15, 153.37, 143.76, 140.67, 137.25, 135.66, 128.75, 128.38, 126.90, 126.63, 125.08, 120.10, 118.69, 37.75, 35.95, 33.81, 30.58, 28.49, and 24.25; HRMS *m*/*z*: 401.2119 [M + H^+^].

Compound(**7i**): *4-(5-isopropyl-2-(4-methoxyphenoxy)phenyl)-4-methylpentanoic acid.* White solid; yield 90.4%; m. p. 121.5–123.1 °C; FT-IR (KBr, cm^−1^): 2959, 2872 (C-H), 1711 (C=O), 1506, 1489, 1455, 1421, 1303, 1249, 1224, and 1220 (Ar-C=C); ^1^H NMR (600 MHz, CDCl_3_) δ 7.10 (d, *J* = 2.1 Hz, 1H, C_6_-H), 6.95 (dd, *J* = 8.3, 2.1 Hz, 1H, C_8_-H), 6.91 (d, *J* = 9.0 Hz, 2H, C_13_-H, C_15_-H), 6.85 (d, *J* = 9.1 Hz, 2H, C_12_-H, C_16_-H), 6.64 (d, *J* = 8.3 Hz, 1H, C_9_-H), 3.78 (s, 3H, C_22_-H), 2.85 (dt, *J* = 13.8, 6.9 Hz, 1H, C_17_-H), 2.25–2.21 (m, 2H, C_2_-H), 2.15–2.10 (m, 2H, C_3_-H), 1.42 (s, 6H, C_20_-H, C_21_-H), and 1.22 (d, *J* = 6.9 Hz, 6H, C_18_-H, C_19_-H); ^13^C NMR (151 MHz, CDCl_3_) δ 180.50, 155.38, 154.70, 150.64, 142.72, 136.23, 126.41, 124.83, 120.27, 118.28, 114.81, 55.64, 37.71, 33.67, 30.63, 28.33, and 24.21; HRMS *m*/*z*: 355.1913 [M + H^+^].

Compound (**7j**): *4-(5-isopropyl-2-(p-tolyloxy)phenyl)-4-methylpentanoic acid.* White solid; yield 96.3%; m. p. 113.6–114.8 °C; FT-IR (KBr, cm^−1^): 2959, (C-H), 1701 (C=O), 1506, 1488, 1456, 1413, 1381, 1306, 1232, and 1210 (Ar-C=C); ^1^H NMR (600 MHz, CDCl_3_) δ 7.12–6.94 (m, 3H, C_6_-H, C_13_-H, C_15_-H), 6.89 (d, *J* = 8.3 Hz, 1H, C_8_-H), 6.78 (d, *J* = 6.6 Hz, 2H, C_12_-H, C_16_-H), 6.62 (d, *J* = 8.3 Hz, 1H, C_9_-H), 2.78 (dt, *J* = 13.8, 6.9 Hz, 1H, C_17_-H), 2.23 (s, 3H, C_22_-H), 2.15–2.10 (m, 2H, C_2_-H), 2.07–2.00 (m, 2H, C_3_-H), 1.33 (s, 6H, C_20_-H, C_21_-H), and 1.16 (d, *J* = 6.9 Hz, 6H, C_18_-H, C_19_-H); ^13^C NMR (151 MHz, CDCl_3_) δ 180.57, 155.09, 153.96, 143.11, 136.74, 132.11, 130.13, 126.43, 124.88, 119.24, 118.65, 37.68, 35.85, 33.71, 30.61, 28.38, 24.21, and 20.64; HRMS *m*/*z*: 339.1963 [M + H^+^].

Compound (**7k**): *4-(5-isopropyl-2-(4-(trifluoromethyl)phenoxy)phenyl)-4-methylpentanoic acid.* White solid; yield 98.3%; m. p. 134.3–136.4 °C; FT-IR (KBr, cm−1): 2966, 2876 (C-H), 1709 (C=O), 1490, 1413, 1322, 1215, 1202, 1215, 1134, 1099, and 1064 (Ar-C=C); 1H NMR (600 MHz, CDCl3) δ 7.54 (d, *J* = 8.6 Hz, 2H, C13-H, C15-H), 7.17 (s, 1H, C6-H), 7.03 (dd, *J* = 21.0, 9.4 Hz, 3H, C8-H, C12-H, C16-H), 6.78 (d, *J* = 8.3 Hz, 1H, C9-H), 2.94–2.86 (m, 1H, C17-H), 2.17–2.07 (m, 4H, C2-H, C3-H), 1.37 (s, 6H, C20-H, C21-H), and 1.25 (d, *J* = 6.9 Hz, 6H, C18-H, C19-H); 13C NMR (151 MHz, CDCl3) δ 180.27, 160.57, 152.03, 144.88, 137.84, 127.40–126.82 (m), 125.31, 120.96, 117.68, 37.69, 36.00, 33.83, 30.43, 28.46, and 24.15; HRMS *m*/*z*: 393.1682 [M + H^+^].

Compound (**7l**): *4-(2-(3-cyanophenoxy)-5-isopropylphenyl)-4-methylpentanoic acid.* White solid; yield 97.7%; m. p. 115.7–117.5 °C; FT-IR (KBr, cm−1): 2965, 2933, 2867 (C-H), 2225 (C≡N), 1709 (C=O), 1586, 1482, 1428, 1412, 1320, 1252, 1225, and 1205 (Ar-C=C); 1H NMR (600 MHz, CDCl3) δ 7.39 (t, *J* = 7.9 Hz, 1H, C14-H), 7.32 (d, *J* = 7.3 Hz, 1H, C13-H), 7.20 (dd, *J* = 16.6, 9.8 Hz, 3H, C6-H, C12-H, C16-H), 7.05 (d, *J* = 8.2 Hz, 1H, C8-H), 6.74 (d, *J* = 8.2 Hz, 1H, C9-H), 2.94–2.84 (m, 1H, C17-H), 2.12 (s, 4H, C2-H, C3-H), 1.37 (s, 6H, C20-H, C21-H), and 1.26 (d, *J* = 6.9 Hz, 6H, C18-H, C19-H); 13C NMR (151 MHz, CDCl3) δ 180.13, 158.20, 151.89, 145.08, 137.82, 130.64, 127.01, 126.01, 125.46, 122.70, 120.88, 120.65, 118.35, 113.48, 37.68, 36.10, 33.82, 30.46, 28.43, and 24.14; HRMS *m*/*z*: 350.1759 [M + H^+^].

Compound (**7m**): *3-(2-(4-carboxy-2-methylbutan-2-yl)-4-isopropylphenoxy)benzoic acid.* White solid; yield 95.8%; m. p. 152.5–155.7 °C; FT-IR (KBr, cm^−1^): 2965, 2869 (C-H), 1716 (C=O), 1589, 1485, 1446, 1414, 1300, 1281, and 1226 (Ar-C=C); ^1^H NMR (600 MHz, CDCl_3_) δ 7.75 (d, *J* = 7.7 Hz, 1H, C_14_-H), 7.46 (d, *J* = 37.9 Hz, 1H, C_16_-H), 7.42 (t, *J* = 7.9 Hz, 1H, C_13_-H), 7.31 (d, *J* = 9.9 Hz, 1H, C_8_-H), 7.15 (d, *J* = 1.9 Hz, 1H, C_6_-H), 7.02 (d, *J* = 10.3 Hz, 1H, C_12_-H), 6.71 (d, *J* = 8.2 Hz, 1H, C_9_-H), 2.89 (dt, *J* = 13.8, 6.9 Hz, 1H, C_18_-H), 2.14 (d, *J* = 7.0 Hz, 2H, C_2_-H), 2.10 (d, *J* = 9.7 Hz, 2H, C_3_-H), 1.41 (d, *J* = 25.1 Hz, 6H, C_21_-H, C_22_-H), and 1.26 (d, *J* = 6.9 Hz, 6H, C_19_-H, C_20_-H); ^13^C NMR (151 MHz, CDCl_3_) δ 180.93, 171.77, 157.66, 152.87, 144.21, 137.09, 131.11, 129.81, 127.04, 125.41, 124.43, 120.01, 119.04, 37.80, 35.47, 33.76, 30.46, 28.51, and 24.16; HRMS *m*/*z*: 369.1705 [M + H^+^].

Compound (**7n**): *4-(2-(3-fluorophenoxy)-5-isopropylphenyl)-4-methylpentanoic acid.* White solid; yield 97.4%; m. p. 112.7–113.9 °C; FT-IR (KBr, cm^−1^): 2962, 2929, 2876 (C-H), 1704 (C=O), 1597, 1482, 1457, 1413, 1305, 1211, and 1119 (Ar-C=C); ^1^H NMR (600 MHz, CDCl_3_) δ 7.25–7.21 (m, 1H, C_13_-H), 7.14 (d, *J* = 2.2 Hz, 1H, C_6_-H), 7.03 (d, *J* = 6.1 Hz, 1H, C_8_-H), 6.78 (d, *J* = 8.3 Hz, 1H, C_9_-H), 6.73 (td, *J* = 7.9, 2.3 Hz, 2H, C_14_-H, C_16_-H), 6.66 (dt, *J* = 10.4, 2.3 Hz, 1H, C_12_-H), 2.88 (dt, *J* = 13.8, 6.9 Hz, 1H, C_17_-H), 2.12 (dt, *J* = 18.3, 9.1 Hz, 4H, C_2_-H, C_3_-H), 1.38 (s, 6H, C_20_-H, C_21_-H), and 1.25 (d, *J* = 6.9 Hz, 6H, C_18_-H, C_19_-H); ^13^C NMR (151 MHz, CDCl_3_) δ 180.22, 164.38, 162.75, 152.53, 144.37, 137.50, 130.39 (d, *J* = 9.8 Hz), 126.71, 125.17, 120.57, 113.71 (d, *J* = 2.9 Hz), 109.34, 109.20, 105.77, 105.61, 37.68, 36.00, 33.79, 30.48, 28.43, and 24.17; HRMS *m*/*z*: 343.1712 [M + H^+^].

Compound (**7o**): *4-(2-(4-chloro-3-(trifluoromethyl)phenoxy)-5-isopropylphenyl)-4-methylpentanoic acid.* White solid; yield 96.7%; m. p. 113.0–114.1 °C; FT-IR (KBr, cm^−1^): 2962, 2927 (C-H), 1711 (C=O), 1477, 1427, 1328, 1164, 1126, 1108, and 1082 (Ar-C=C); ^1^H NMR (600 MHz, CDCl_3_) δ 7.39 (d, *J* = 8.8 Hz, 1H, C_13_-H), 7.31 (d, *J* = 2.9 Hz, 1H, C_16_-H), 7.16 (d, *J* = 2.1 Hz, 1H, C_6_-H), 7.05–7.01 (m, 2H, C_8_-H, C_12_-H), 6.72 (d, *J* = 8.3 Hz, 1H, C_9_-H), 2.89 (dt, *J* = 13.8, 6.9 Hz, 1H, C_17_-H), 2.15–2.09 (m, 4H, C_2_-H, C_3_-H), and 1.37 (s, 6H, C_20_-H, C_21_-H), 1.25 (d, *J* = 6.9 Hz, 6H, C_18_-H, C_19_-H); ^13^C NMR (151 MHz, CDCl_3_) δ 180.03, 156.26, 152.19, 144.89, 137.57, 132.57, 126.97, 125.41, 121.95, 120.18, 117.46 (d, *J* = 5.4 Hz), 37.67, 35.99, 33.77, 30.39, 28.38, and 24.10; HRMS *m*/*z*: 427.1289 [M + H^+^].

Compound(**7p**): *4-(2-(3,4-difluorophenoxy)-5-isopropylphenyl)-4-methylpentanoic acid.* White solid; yield 98.8%; m. p. 106.4–108.2 °C; FT-IR (KBr, cm^−1^): 3061 (Ar-H), 2962, 2929, 2874 (C-H), 1702 (C=O), 1513, 1489, 1456, 1412, 1303, 1249, 1213, 1145, and 1102 (Ar-C=C); ^1^H NMR (600 MHz, CDCl_3_) δ 7.14 (d, *J* = 2.2 Hz, 1H, C_13_-H), 7.10–7.05 (m, 1H, C_6_-H), 7.02 (d, *J* = 8.3 Hz, 1H, C_8_-H), 6.79 (dd, *J* = 11.5, 6.7 Hz, 1H, C_9_-H), 6.72 (d, *J* = 8.3 Hz, 1H, C_16_-H), 6.69–6.65 (m, 1H, C_12_-H), 2.87 (dd, *J* = 13.8, 6.9 Hz, 1H, C_17_-H), 2.14 (d, *J* = 12.1 Hz, 2H, C_2_-H), 2.12–2.08 (m, 2H, C_3_-H), 1.38 (s, 6H, C_20_-H, C_21_-H), and 1.24 (d, *J* = 6.9 Hz, 6H, C_18_-H, C_19_-H; ^13^C NMR (151 MHz, CDCl_3_) δ 180.06, 152.89, 144.35, 137.27, 126.79, 125.23, 119.95, 117.44 (d, *J* = 19.6 Hz), 113.77 (dd, *J* = 5.8, 3.5 Hz), 107.87, 107.74, 37.69, 35.97, 33.77, 30.45, 28.39, and 24.16; HRMS *m*/*z*: 361.1619 [M + H^+^].

Compound (**7q**): *4-(2-(4-chlorophenoxy)-5-isopropylphenyl)-4-methylpentanoic acid.* White solid; yield 90.8%; m. p. 121.5–123.8 °C; FT-IR (KBr, cm^−1^): 2956, 2871 (C-H), 1709 (C=O), 1483, 1456, 1411, 1291, 1244, 1197, and 1082 (Ar-C=C); ^1^H NMR (600 MHz, CDCl_3_) δ 7.25 (d, *J* = 9.1 Hz, 2H, C_13_-H, C_15_-H), 7.13 (d, *J* = 2.2 Hz, 1H, C_6_-H), 7.00 (d, *J* = 6.2 Hz, 1H, C_8_-H), 6.89 (d, *J* = 8.9 Hz, 2H, C_12_-H, C_16_-H), 6.71 (d, *J* = 8.3 Hz, 1H, C_9_-H), 2.87 (dt, *J* = 13.8, 6.9 Hz, 1H, C_17_-H), 2.20–2.14 (m, 2H, C_2_-H), 2.13–2.08 (m, 2H, C_3_-H), 1.38 (s, 6H, C_20_-H, C_21_-H), and 1.24 (d, *J* = 6.9 Hz, 6H, C_18_-H, C_19_-H); ^13^C NMR (151 MHz, CDCl_3_) δ 180.00, 156.22, 153.05, 144.04, 137.25, 129.60 (d, *J* = 4.9 Hz), 127.53, 126.68, 125.12, 119.93, 119.59, 37.68, 35.93, 33.76, 30.46, 28.40, and 24.17; HRMS *m*/*z*: 359.1418 [M + H^+^].

Compound (**7r**): *4-(5-isopropyl-2-(3-methoxyphenoxy)phenyl)-4-methylpentanoic acid.* White solid; yield 94.2%; m. p. 125.6–126.9 °C; FT-IR (KBr, cm^−1^): 2959, 2874 (C-H), 1708 (C=O), 1603, 1486, 1454, 1412, 1386, 1282, 1214, 1162, 1141, 1083, and 1043 (Ar-C=C); ^1^H NMR (600 MHz, CDCl_3_) δ 7.17 (t, *J* = 8.1 Hz, 1H, C_13_-H), 7.13 (s, 1H, C_6_-H), 6.99 (d, *J* = 8.3 Hz, 1H, C_8_-H), 6.77 (d, *J* = 8.3 Hz, 1H, C_9_-H), 6.59 (d, *J* = 8.7 Hz, 1H, C_16_-H), 6.53 (d, *J* = 8.7 Hz, 2H, C_12_-H, C_14_-H), 3.75 (s, 3H, C_22_-H), 2.90–2.84 (m, 1H, C_17_-H), 2.22–2.17 (m, 2H, C_2_-H), 2.12 (dd, *J* = 9.1, 7.3 Hz, 2H, C_3_-H), 1.40 (s, 6H, C_20_-H, C_21_-H), and 1.24 (d, *J* = 7.0 Hz, 6H, C_18_-H, C_19_-H); ^13^C NMR (151 MHz, CDCl_3_) δ 180.83, 160.92, 158.76, 153.18, 143.66, 137.11, 130.02, 126.51, 125.00, 120.14, 110.67, 108.30, 104.54, 60.45, 55.27, 38.10, 35.94, 33.95–33.56, 30.64, 28.43, 24.13, and 14.15; HRMS *m*/*z*: 355.1912 [M + H^+^].

Compound (**7s**): *4-(2-(2-chlorophenoxy)-5-isopropylphenyl)-4-methylpentanoic acid.* White solid; yield 94.9%; m. p. 124.1–125.8 °C; FT-IR (KBr, cm^−1^): 3432 (Ar-H), 2959, 2925, 2870 (C-H), 1708 (C=O), 1475, 1446, 1412, 1263, 1238, 1200, 1082, and 1059 (Ar-C=C); ^1^H NMR (600 MHz, CDCl_3_) δ 7.44 (d, *J* = 6.4 Hz, 1H, C_15_-H), 7.18–7.12 (m, 2H, C_6_-H, C_13_-H), 7.02 (t, *J* = 7.7 Hz, 1H, C_14_-H), 6.98 (d, *J* = 8.3 Hz, 1H, C_8_-H), 6.91 (d, *J* = 9.6 Hz, 1H, C_12_-H), 6.59 (d, *J* = 8.3 Hz, 1H, C_9_-H), 2.87 (dt, *J* = 13.8, 6.9 Hz, 1H, C_17_-H), 2.26–2.21 (m, 2H, C_2_-H), 2.16–2.11 (m, 2H, C_3_-H), 1.44 (s, 6H, C_20_-H, C_21_-H), and 1.23 (d, *J* = 6.9 Hz, 6H, C_18_-H, C_19_-H); ^13^C NMR (151 MHz, CDCl_3_) δ 180.83, 160.92, 158.76, 153.18, 143.66, 137.11, 130.02, 126.51, 125.00, 120.14, 110.67, 108.23, 104.54, 55.27, 37.67, 35.94, 33.74, 30.64, 28.43, and 24.19; HRMS *m*/*z*: 359.1418 [M + H^+^].

Compound (**7t**): *4-(5-isopropyl-2-(pyridin-2-yloxy)phenyl)-4-methylpentanoic acid.* White solid; yield 97.6%; m. p. 133.2–134.9 °C; FT-IR (KBr, cm^−1^): 2960, 2926, 2871 (C-H), 1710 (C=O), 1574, 1490, 1476, 1427, 1245, 1200, and 1081 (Ar-C=C); ^1^H NMR (600 MHz, CDCl_3_) δ 8.20 (d, *J* = 6.4 Hz, 1H, C_15_-H), 7.65 (t, *J* = 7.8 Hz, 1H, C_13_-H), 7.16 (s, 1H, C_6_-H), 7.06 (d, *J* = 10.3 Hz, 1H, C_8_-H), 6.96–6.93 (m, 1H, C_9_-H), 6.88–6.84 (m, 2H, C_12_-H, C_14_-H), 2.89 (dt, *J* = 13.8, 6.9 Hz, 1H, C_16_-H), 2.11 (s, 4H, C_2_-H, C_3_-H), and 1.37 (s, 6H, C_19_-H, C_20_-H), 1.25 (d, *J* = 7.0 Hz, 6H, C_17_-H, C_18_-H); ^13^C NMR (151 MHz, CDCl_3_) δ 179.64, 163.70, 150.62, 147.75, 144.80, 139.56, 137.90, 126.67, 125.07, 122.52, 118.27, 111.71, 37.63, 36.33, 33.83, 30.53, 28.58, and 24.16; HRMS *m*/*z*: 326.1759 [M + H^+^].

Compound (**7u**): *4-(5-isopropyl-2-(pyridin-3-yloxy)phenyl)-4-methylpentanoic acid.* White solid; yield 98.2%; m. p. 131.5–132.6 °C; FT-IR (KBr, cm^−1^): 2960, 2926, 2871 (C-H), 1710 (C=O), 1574, 1490, 1476, 1427, 1245, 1200, and 1081 (Ar-C=C); ^1^H NMR (600 MHz, CDCl_3_) δ 8.31 (d, *J* = 23.2 Hz, 2H, C_14_-H, C_15_-H), 7.33 (s, 1H, C_12_-H), 7.29 (s, 1H, C_6_-H), 7.17 (s, 1H, C_13_-H), 7.02 (d, *J* = 10.2 Hz, 1H, C_8_-H), 6.71 (d, *J* = 8.3 Hz, 1H, C_9_-H), 2.92–2.85 (m, 1H, C_16_-H), 2.13 (dd, *J* = 26.7, 10.2 Hz, 4H, C_2_-H, C_3_-H), 1.40 (s, 6H, C_19_-H, C_20_-H), and 1.24 (d, *J* = 6.9 Hz, 6H, C_17_-H, C_18_-H); ^13^C NMR (151 MHz, CDCl_3_) δ 178.01, 152.47, 144.67, 142.74, 139.96, 137.68, 127.03, 126.08, 125.27, 124.45, 119.79, 37.82, 36.32, 33.79, 30.74, 29.70, 28.44, and 24.16; HRMS *m*/*z*: 326.1760 [M + H^+^].

Compound (**7v**): *4-(5-isopropyl-2-(quinolin-7-yloxy)phenyl)-4-methylpentanoic acid.* White solid; yield 97.4%; m. p. 136.8–137.2 °C; FT-IR (KBr, cm^−1^): 2960, 2871 (C-H), 1710 (C=O), 1504, 1489, 1463, 1324, 1259, 1225, 1204, and 1081 (Ar-C=C); ^1^H NMR (600 MHz, CDCl_3_) δ 8.62 (s, 1H, C_16_-H), 8.01 (t, *J* = 7.9 Hz, 2H, C_12_-H, C_14_-H), 7.50 (dd, *J* = 9.2, 1.7 Hz, 1H, C_18_-H), 7.28–7.26 (m, 1H, C_15_-H), 7.21 (d, *J* = 12.4 Hz, 2H, C_6_-H, C_11_-H), 7.06 (d, *J* = 8.3 Hz, 1H, C_8_-H), 6.84 (d, *J* = 8.1 Hz, 1H, C_9_-H), 2.96–2.90 (m, 1H, C_20_-H), 2.23–2.19 (m, 2H, C_2_-H), 2.18–2.12 (m, 2H, C_3_-H), 1.45 (s, 6H, C_23_-H, C_24_-H), and 1.28 (d, *J* = 6.9 Hz, 6H, C_21_-H, C_22_-H); ^13^C NMR (151 MHz, CDCl_3_) δ 177.98, 156.11, 152.48, 147.66, 144.60, 143.60, 138.00, 136.01, 129.98, 129.31, 127.06, 125.1, 123.56, 121.34, 120.83, 112.02, 37.91, 36.35, 33.84, 30.94, 28.56, and 24.21; HRMS *m*/*z*: 376.1916 [M + H^+^].

### 3.3. Antifungal Activity Test

And all of the plant pathogens used in the test were gained from the Biological Assay Center, Nankai University, Tianjin, China. The test reagent was dissolved in acetone and diluted into a 500 ppm solution with a 200 ppm SorporL-144 emulsifier. Then, 1 mL of the drug solution was taken and injected into the Petri dish, and 9 mL PSA medium was added to make the final concentration of 50 ppm drug-containing plate. The culture plates were incubated at 24 ± 1 ℃, and the extended diameters of the mycelium circles were calculated after 48 h. Finally, the inhibitory percentages of all compounds tested were measured by comparing the mycelium diameter of the fungi to the blank control.

### 3.4. 3D-QSAR Analysis

For further study of the relationship between the structures of the target compounds’ antifungal activities and their substituents, the 3D-QSAR model was built applying the CoMFA pattern of Sybyl-X2.1.1 software [40]. According to the reference [41], the structures of compounds **7a–7s** were optimized based on the Gasteiger–Hückel charges and Tripos force field. Compound **7b** with the best activity was applied as the template molecule and the common skeleton atoms are marked with an asterisk, as shown in Figure 7.

The nineteen target compounds were superimposed, and the result is shown in Figure 8. The inhibition rate against *A. solani* was converted to the AF using the formula: AF = log{[relative inhibitory rate/(100 − relative inhibitory rate)] × molecular weight}. The established 3D-QSAR model was inspected by the partial least-squares means. Its predictive ability was estimated by a cross-validated value squared (*q*^2^), a correlation coefficient squared (*r*^2^), a standard deviation (*S*), and a Fisher validation value (*F*).

### 3.5. Preparation of Nano **MgAl-LDH** Carrier

The **MgAl-LDH** was synthesized by a “bottom-up” method [21]. Specific methods were as follows: Solution A: a mixture of Mg(NO_3_)_2_∙6H_2_O (0.103 g, 0.4 mmol) and Al(NO_3_)_3_∙9H_2_O (0.075 g 0.2 mmol) dissolved in 40 mL of deionized water. Solution B: 40 mL aqueous solution containing 25% formamide and NaNO_3_ (0.017 g, 2 mmol). Solution C: a solution of NaOH (0.180 g, 4.5 mmol) in water (30 mL). A and C were slowly added to B under the condition of stirring at 80 °C, and the pH of the mixture was detected by pH test paper and kept at about 9. After dropping the solution, continue to stir for 30 min to make it fully nucleated and crystallized, get white colloid, centrifuge, and then wash with deionized water and ethanol to precipitate three times each. The toxic formamide was then removed by alternate dialysis and centrifugation, and the residues were dispersed into deionized water for later use.

### 3.6. Preparation and In Vitro pH Controlled-Releasing Evaluation of Drug-Loading Complexes

The drug-loaded complex was prepared by using compound **7b** as the representative. Firstly, compound **7b** was dissolved in 50% ethanol solution, and then, **MgAl-LDH** and compound **7b** were mixed in a mass ratio of 1:1 and stirred for 6 h at room temperature. After that, the mixture was centrifuged at 9000 rpm for 6 min, and the supernatant was removed. The precipitate was washed with deionized water three times and dried in an oven at 40 °C to obtain the drug-loading complexes **7b/MgAl-LDH-1**. Similarly, the mass ratios of compound **7b** and **MgAl-LDH** in the mixtures were changed into 2:1 and 3:1, and the above-mentioned steps were repeated to obtain complexes **7b/MgAl-LDH-2** and **7b/MgAl-LDH-3**, respectively.

Then the in vitro controlled-releasing performance of the **7b/MgAL-LDH-2** complex was evaluated. 5.0 mg of the drug complex was placed in 50 mL of ethanol-aqueous solution (1:10, v/v) under different pH values (5.7, 7.0, and 8.2) at room temperature, and 4 mL of the clarified solution was taken from the whole system at a specific time point, then also detected by UV-1800 spectrophotometer. The cumulative releasing percentage (%) was calculated according to the formula: cumulative releasing percentage (%) = cumulative releasing amount of drug/total drug-loaded of a complex sample, and the cumulative releasing percentage (%) was calculated to obtain the spectra of various complexes.

### 3.7. Characterization

In the first place, intermediate **2–4** were characterized according to our previously reported research [27,28]. Then, IR spectra of compound **5**, **6a~6v**, **7a~7v**, carrier **MgAl -LDH**, and complex **7b/MgAl-LDH-1**, **7b/MgAl-LDH-2**, and **7b/MgAl-LDH-3** were measured in a Nicolet iS50 FT-IR spectrometer (Thermo Scientific Co., Ltd., Madison, WI, USA) using KBr pellets. A Bruker Avance III HD 500 MHz/600MHz spectrometer was used to measure ^1^H and ^13^C NMR spectra of compound **5**, **6a-6v**, **7a-7v** using CDCl_3_ as the solvent, and TMS as an internal standard. Moreover, HRMS spectra of compounds **5**, **6a-6v**, and **7a-7v** were detected on Thermo Scientific Q Exactive instrument with atmospheric pressure chemical ionization source (APCI). And melting points of compounds **5**, **6a-6v**, and **7a-7v** were recorded on an MP420 automatic melting point apparatus (Hanon Instruments Co., Ltd., Jinan, China) and have proved correct. Besides, the high-performance liquid chromatography (HPLC) analysis of compounds **5**, **6a-6v**, and **7a-7v** was performed on a Waters 1525 instrument (Waters, Milford, MA, USA) equipped with chromatographic column SunFire C18 (4.6 × 150 mm) and 2998 PDA detector.

Next, for the fabrication of carriers and drug-loading complexes, the ultrasonic treatment was carried out with an XO-SM50 ultrasonic microwave reaction system (Nanjing Xianou Instrument Manufacturing Co., Ltd., Nanjing, China). UV spectra were measured by Shimadzu UV-1800 spectrophotometer. The wide-angle X-ray diffraction (XRD) analysis of carrier **MgAl-LDH** and complex **7b/MgAl-LDH-2** was carried out on Bruker D8 Advance with Cu Kα radiation (1.54 Å) at 40 kV and 40 mA, and the patterns were recorded in 2*θ* = 5~90°. The scanning electron micrograph (SEM) and atomic force microscope (AFM) images of carrier **MgAl-LDH** and complex **7b/MgAl-LDH-2** were imaged using Zeiss Sigma 300 and Dimension Edge, respectively.

## 4. Conclusions

Twenty-two diphenyl ether-carboxylic acid compounds **7a–7v** were successfully synthesized from longifolene and characterized by FT-IR, ^1^H NMR, ^13^C NMR, and HRMS. The results of the preliminary bioassay showed that compound **7b** displayed significant and broad-spectrum antifungal activity. Meanwhile, a reasonable and efficient 3D-QSAR model has been established. The drug carried complexes **7b/MgAl-LDH** were prepared using compound **7b** and Mg-Al hydrotalcite and characterized. Meanwhile, their controlled release behavior was investigated. As a result, complex **7b/MgAl-LDH-2** exhibited excellent controlled-releasing performance in the water/ethanol (10:1, v:v) and under a pH of 5.7. In conclusion, compound **7b**, with significant and broad-spectrum antifungal activity, and its nano complexes of **MgAl-LDH**, possessing excellent pH-controlled-releasing performance, could serve as the leading complexes for further investigation.

## Data Availability

Data is contained within the article or the supplementary data file.

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
