# Peer review of "Synthesis, Antifungal Activity, 3D-QSAR and Controlled Release on Hydrotalcite Study of Longifolene-Derived Diphenyl Ether Carboxylic Acid Compounds"

_molecules, 2023, doi:10.3390/molecules28041911_

Round 1

Reviewer 1 Report

1. At 2.2 heading last line....Chlolothalonil mentioned.

But at in table 1. Reference drug name is different. Which is correct?

2. Compare to reference drug ..the values are almost double in vase of 7b against p. Pirocola fungi. That means your compound is highly active than drug reference right? 

3. 7l compound also same. Other compounds also showed high values than reference. Which how you explain ? Appropriately 

Author Response

  1. At 2.2 heading last line....Chlolothalonil mentioned. But at in table 1. Reference drug name is different. Which is correct?

Answer: Thank you for your reminding. At 2.2 heading last line....Chlolothalonil mentioned is correct, the one in table 1 is a writing error. The related content has been revised (see table 1 in page 4).

  1. Compared to reference drug, the values are almost double in vase of 7b against p. Pirocola fungi. That means your compound is highly active than drug reference right?
  2. 7l compound also same. Other compounds also showed high values than reference. Which how you explain ? Appropriately

Answer: Thank you for your advice. Although some target compounds showed higher values than reference chlorothalonil against P. pirocola fungi, the activity of reference chlorothalonil was relatively weak against P. pirocola fungi. We therefore did not discuss this point.

In the preliminarily screening stage we cannot explain the reason. Further studies will be carried out on basis of the result reported in this paper.

Reviewer 2 Report

The set of compounds 7a-7v, were synthesized from longifolene and characterized using FT-IR, 1H NMR, 13C NMR, and HRMS techniques. The preliminary bioassay results showed that compound 7b exhibited strong and broad-spectrum antifungal activity. Additionally, a 3D-QSAR model was developed. The drug-carrier complexes 7b/MgAl-LDH were then prepared using compound 7b and Mg-Al hydrotalcite and characterized. The controlled-release behavior of these complexes was also investigated. As a result, complex 7b/MgAl-LDH-2 demonstrated outstanding controlled-release performance in a water/ethanol (10:1, v:v) system under a pH of 5.7.

This work should be improved according to the following comments:

1) L.84: Scheme 1. Yields should be added for all products.

2) L.92: The discussion about NMR spectra is not informative and does not allow to confirm the structure of these compounds. Please provide more detail or refrain from discussing.

3) Did you use your 3D-QSAR model to predict a new active compound of this type, synthesize it, and test it in vitro against plant pathogens?

4) In the section “3.2. Chemical Synthesis”. You should add yields for compounds 5 and 6a-6v. Also add a picture of the compound 7 scaffold with atom numbering according to the 1H NMR description.

5) In the section “3.6. Preparation and in vitro pH controlled-releasing evaluation of drug-loading complexes”. What was the oven temperature used? Did you check the supernatant for residues of the substance? Can you explain why you used 7b/MgAL-LDH-2? Was the in vitro controlled-releasing performance of the 7b/MgAL-LDH-1 and 7b/MgAL-LDH-3 complexes evaluated?

6) In supplementary material: Correct the scale on the f1 axis to fit spectrum size for all 1H NMR spectra.

Author Response

1) L.84: Scheme 1. Yields should be added for all products.

Answer: Thank you for your advice. The yields of all compounds have been added. The related contents have been revised in the section “3.2. Chemical Synthesis” (see line 238 in page 9; line 240 in page 9; see lines 250-251 in page 9; lines 262-263 in page 9)

2) L.92: The discussion about NMR spectra is not informative and does not allow to confirm the structure of these compounds. Please provide more detail or refrain from discussing.

Answer: Thank you for your suggestion. The related contents have been provided on the paper (see lines 92-100 in page 3).

3) Did you use your 3D-QSAR model to predict a new active compound of this type, synthesize it, and test it in vitro against plant pathogens?

Answer: Thank you for your advice. We have used the 3D-QSAR model to predict a new active compound of this type, and the related contents have been added to the paper (see lines 158-161 in page 6; see line167 in figure 3).

This work is only in the preliminarily screening stage with the purpose of seeking for leading compound worthy of deep research. These proposed compounds will be taken into consideration in the further study.

4) In the section “3.2. Chemical Synthesis”. You should add yields for compounds 5 and 6a-6v. Also add a picture of the compound 7 scaffold with atom numbering according to the 1H NMR description.

Answer: Thank you for your advice. The related yields have been added on the paper (see lines 250-251 in page 9; lines 262-263 in page 9). The atom numbering of the compound 7 according to the 1H NMR description has been added in the supplementary (see the supplementary file).

5) In the section “3.6. Preparation and in vitro pH controlled-releasing evaluation of drug-loading complexes”. What was the oven temperature used? Did you check the supernatant for residues of the substance? Can you explain why you used 7b/MgAL-LDH-2? Was the in vitro controlled-releasing performance of the 7b/MgAL-LDH-1 and 7b/MgAL-LDH-3 complexes evaluated?

Answer: Thank you for this enlightening comment. The oven temperature used was 40 °C, and the related content has been added (see line 522, page 15). The drug-loaded complexes 7b/MgAL-LDH-1, 7b/MgAL-LDH-2, and 7b/MgAL-LDH-3 were prepared by mixing MgAl-LDH with compound 7b in different mass ratios. However, the drug-loading capacity of 7b/MgAL-LDH-2 was better than that of 7b/MgAL-LDH-1 and 7b/MgAL-LDH-3. Therefore, complex 7b/MgAL-LDH-2 was selected in our study on the controlled-releasing performance. These comments will be taken into consideration in the further study.

6) In supplementary material: Correct the scale on the f1 axis to fit spectrum size for all 1H NMR spectra.

Answer: Thanks for this enlightening comment. All 1H NMR spectra have been corrected on the f1 axis to fit spectrum size (see the supplementary file).
